# Validation of the QualiPresc instrument for assessing the quality of drug prescription writing in primary health care

**Almária Mariz Batista** [1,2¤]*, **Zenewton André da Silva Gama**[3], **Dyego Souza**[2,3]

**1** Escola Multicampi de Ciências Médicas, Universidade Federal do Rio Grande do Norte, Caicó, Brazil,
**2** Programa de Pós-Graduação em Ciências da Saúde, Universidade Federal do Rio Grande do Norte, Natal,
Brazil, **3** Departamento de Saúde Coletiva, Universidade Federal do Rio Grande do Norte, Natal, Brazil

¤ Current address: Universidade Federal do Rio Grande do Norte, Escola Multicampi de Ciências Médicas
do Rio Grande do Norte, Caicó, Rio Grande do Norte, Brazil
* almariamariz@yahoo.com.br

## Abstract

### Introduction

Adverse events related to drug prescriptions are the main patient safety issue in primary
care; however there is a lack of validated instruments for assessing the quality of prescrip-
tion writing, which covers the prescriber, the patient and the drug information.

### Objective

To develop and validate the QualiPresc instrument to assess and monitor the quality of drug
prescriptions in primary care, accompanied by a self-instruction direction, with the goal of fill-
ing the gap in validated instruments to assess the quality of prescription writing.

### Methodology

A validation study conducted in a municipality in Northeastern Brazil, based on prescriptions
prepared in January 2021 by physicians assigned to 18 Basic Health Units and filed in 6 dis-
tribution/dispensing units. Four steps were covered: 1) Analysis of content validity of each
indicator (relevance and adequacy); 2) Analysis of reliability via intra and inter-rater agree-
ment of each indicator; 3) Analysis of the utility of each indicator; 4) Construction and analy-
sis of the reliability of a weighted composite indicator based on effectiveness and safety
scores for each indicator.

### Results

Twenty-nine potential indicators were listed, but only 13 were approved for validity, reliability
and usefulness. Twelve indicators were excluded because of validity (<90% validity index)
and four because they were not useful in the context of the study. Three weighted composite
indicators were tested, but only one was approved for reliability and usefulness. The vali-
dated instrument therefore contains 13 indicators and 1 weighted composite indicator.

pone.0267707

de Minas Gerais, BRAZIL

**Data Availability Statement:** All relevant data are
within the paper and its Supporting Information
files.

**Funding:** This study received financial support from the Coordenação de Aperfeiçoamento de Pessoal de Nível Superior (CAPES). The funders had no role in study design, data collection and analysis, decision to publish, or preparation of the manuscript.

**Competing interests:** The authors have declared that no competing interests exist.

## Conclusion

This study demonstrates the validity, reliability and usefulness of QualiPresc for the evaluation of prescription writing in the context of primary care. Application to contexts such as secondary care and tertiary care requires cross-cultural adaptation and new content validity. Educators, managers and health care professionals can access QualiPresc online, free of charge, to assess performance and provide feedback involving drug prescribers.

## Introduction

Unsafe medication processes occur in all care settings, but they weigh more heavily in primary care. In this context, much of the harm related to health care seems to be associated with drugs prescription and with errors/delays in diagnosis and its avoidability is related more to medication, administrative practices and communication [1]. Further studies in primary care have found medication to be a critical point of patient safety [2–5]. Globally, up to 40% of patients suffer harm related to health care in the context of outpatient care and up to 80% of these are preventable [6].

Primary care has peculiarities that can make the medication process unsafe, but studies in this environment are less frequent than in the hospital environment [7]. There is a need for interventions related to information technology infrastructure, educational support for prescription management, clinical pharmacy services, medication reconciliation and strategies for measuring quality indicators [8].

In the medication process, prescription stand out as the step most susceptible to medication error [9–11]; therefore, they are a priority for the measurement of quality indicators. About 40% of users assisted in primary care do not need medication, however, 80% of these leave consultations with a prescription [12–14].

In the context of primary care, drugs demand/supply is high and technological advances are fast, which can lead to less than adequate prescription practices [15], due to the commitment and time investment demanded from prescribers in order to gain the appropriate knowledge inherent to this health technology, including the best scientific evidence about efficacy, safety and cost-effectiveness. This was evidenced in the Covid-19 pandemic, which required an investment of time by prescribers to obtain appropriate knowledge about the efficacy, safety and cost-effectiveness of prescribed drugs [16].

The preparation of a drug prescription involves structuring a document based on the technical specifications of the drug (prescription writing), following the prescriber's therapeutic decision (therapeutic decision criteria). Ideal prescription writing encompasses the prescriber, the patient and the drug information; however, this information is often missing or incomplete [17–25]. In Brazil, the practice of prescription writing is based on recommendations from the Ministry of Health [26].

The quality of a prescription is reflected across various dimensions of the quality of health care, in this case, the safety, effectiveness, cost and patient-centered care, which results in improved care through user satisfaction, lower costs and a reduction of medication errors [27].

Thus, in order to reduce medication errors and reach the WHO global goal, Medication Without Harm [28], it is essential to evaluate and improve the quality of drug prescriptions. Failure to observe these aspects during the preparation of a prescription results in prescription error, a type of medication error related to decision-making or writing which, although unintentional, can reduce the efficacy of the treatment or increase the risk of mistakes [29].

Despite the abundance of studies on prescription quality in primary care according to therapeutic decision criteria [30–35], there is no valid, reliable and useful set of indicators that allows the assessment of prescription writing quality. The absence of a set of indicators to assess the quality of prescription writing is an incomprehensible gap, given that medication errors represent the main safety incidents resulting in adverse events in primary care [2–5]. Furthermore, this cannot be resolved through an electronic prescription system, even in well-resourced settings [36]. Therefore, it is extremely necessary to solve this deficiency, especially in primary care. Furthermore, the availability of a set of indicators to assess prescriptions from this perspective can be useful in health services and in teaching good prescribing practices.

Thus, this study aims to develop and validate a concise instrument to monitor the quality of prescription writing, to encourage continuous evaluation and improvement of the effectiveness and safety of pharmacological therapy in primary health care.

## Methodology

### Context and design of the study

This study is contextualized as a research/university extension project to improve the quality of medication prescription in the context of primary care, based on the cooperation between a Brazilian federal university and the health department of a municipality in Northeastern Brazil.

It is a validation study developed from March 2020 to March 2021, involving four steps: 1) Validity analysis; 2) Reliability analysis; 3) Utility analysis; 4) Attribution of scores to the approved indicators and analysis of the reliability of composite indicators. The flow of steps/products is shown in Fig 1.

As this study is a validation study of a measurement instrument for the management of quality in healthcare, it does not meet any of the specifications provided for in the EQUATOR Network, and it is therefore not possible to present the Research Reporting Guideline checklist here.

### Step 1. Validity analysis

The validity analysis covers analysis of the relevance (face validity), adequacy (content validity) and theoretical support (criteria validity) of each potential indicator used to assess prescription quality. In this case, the relevance indicates how important the potential indicator is in assessing the quality of the prescription, the adequacy indicates how much the potential indicator is able to measure the quality of the prescription and the theoretical support refers to the literature on which the potential indicator is based. Thus, potential indicators were listed from a review of national and international literature on recommendations for the prevention of medication errors and safe prescribing practices [26, 27, 37–41]. Consensus on the indicators to integrate the instrument was established via the Delphi panel [42], which evaluated each item for adequacy and relevance as indicators of prescription quality.

Pharmacists, physicians, nurses and dentists with experience of work processes in academia and the health service were invited to participate in the Delphi panel. Thus, it covered a target audience that either evaluates the quality of the prescription and/or has its performance evaluated [40]. Among the 21 invited participants, 15 agreed to participate. The recruitment of specialists took place via telephone contact, during which the methodology of the Delphi panel was presented. After accepting the invitation, the material to be analyzed was sent via e-mail along with the informed consent form.

The Delphi panel was developed in three rounds, for each of which a Likert scale (0–9) was created for both relevance and adequacy of potential indicators. Each potential indicator was

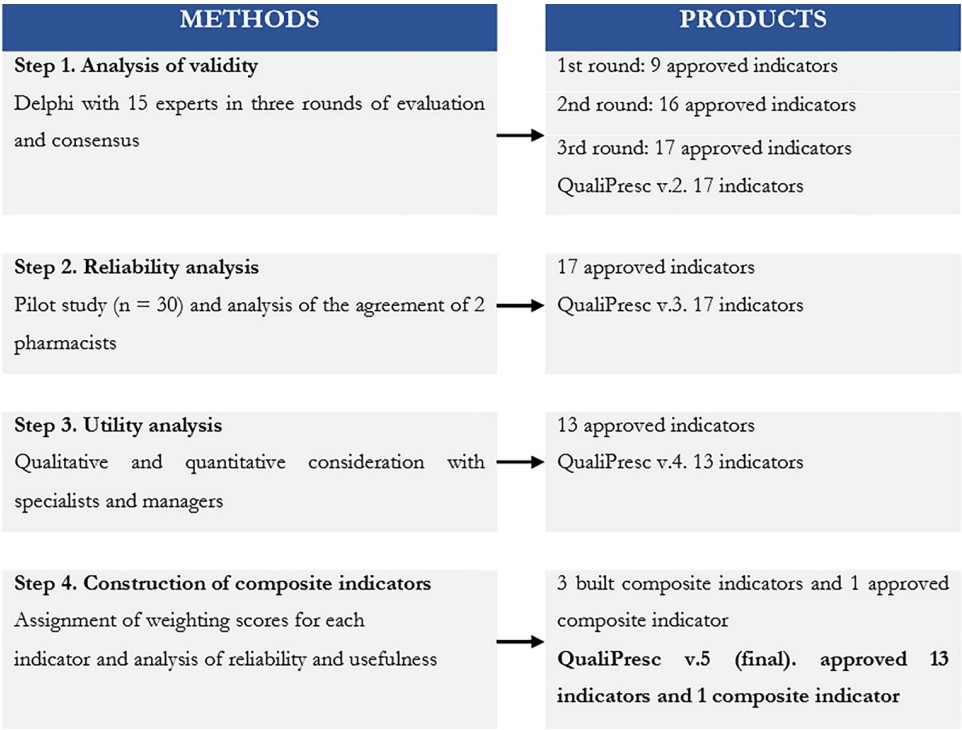

**Fig 1. Flow of steps and products for the development and validation of the QualiPresc.**

considered approved if at least 90% of the values assigned by experts where > 7 on the Likert scale [43], while those considered non-consensual were reassessed in subsequent rounds based on this same system. For the 1st round, specialists were given a period of 30 days to analyze the material, while, a period of 15 days was granted for both the 2nd and 3rd round. At the end of each time period, telephone contact was made with any specialists that in had not completed the analysis, in order to agree on a new delivery period, which contributed to 100% adherence.

In the 1st round, each potential indicator was analyzed as to its relevance, adequacy and theoretical support in assessing the quality of the prescription. The impact of each potential indicator on the safety and efficacy of the drug treatment was also considered in the 1st round, with the product of the score of these two items constituting the score assigned by each specialist. The final score of each potential indicator corresponded to the product of the average efficacy and safety from the 15 experts. For the 2nd and 3rd rounds, only the relevance and adequacy of each potential indicator were analyzed.

## Step 2. Reliability analysis

The inter-rater reliability was assessed by two pharmacists through an independent review of 30 randomly selected drug prescriptions [44], based on duplicate prescriptions and special control prescriptions. These were prepared in January 2021 by 24 physicians assigned to Basic Health Units, and were filed in 5 district pharmacies and 1 pharmaceutical supply center in the municipality of the study.

This Brazilian city has an estimated population of 68,343 inhabitants [45] and, in the context of primary care, has 24 Family Health Teams, distributed in 18 Basic Health Units. It also features five district pharmacies and one pharmaceutical supply center, which centralizes the process of distribution/dispensing of prescription drugs in this context.

Inter-rater agreement was calculated from the % agreement (general agreement index) and Cohen's kappa. The general agreement index is sufficient when agreement > 95%. A negative kappa or 0 indicates that agreement is low or non-existent; >0.2, mild; >0.4, moderate; > 0.6, good; > 0.8, high; and 1 indicates complete agreement. A low kappa (<0.4) should not be interpreted in the case of extreme presence of indicator compliance (<15% or >85%), as it may be biased. In this case, it was planned to intentionally select a sample with unknown extreme presence and recalculate kappa [46]. A kappa cut off point with 95% confidence interval (95% CI) ≥ 0.4 was considered.

### Step 3. Utility analysis

After validating the indicators in terms of validity and reliability, they were analyzed in terms of the usefulness of their application. From this perspective, if the indicator is 100% in compliance with the pilot, it is not useful to monitor. If it is 0% compliant, it is necessary to ask if there is governance to include it in the monitoring. In this case, a pharmacist, a specialist in quality management and the municipal health secretary of that municipality were consulted. Where there was consensus about the usefulness of the indicator, it was considered useful; otherwise, it was excluded.

### Step 4. Attribution of scores to the approved indicators and reliability analysis of the composite indicators

Three composite indicators were created: effectiveness, safety and quality of the prescription. These, made up of heterogeneous and uncorrelated indicators, correspond to a Formative Measurement Model, which advocates that the value of a construct is determined by the values of a set of indicators. In this perspective, the instrument's score resulted from the sum of its items, which were given a weighting that reflected the contribution of each item to the construct [47].

For this purpose, a 3x3 matrix was built combining 2 dimensions, representing the impact of each item on the safety and efficacy of the treatment, with 3 levels each (absent = 1, moderate = 2, significant = 3). The 15 experts defined the values of this matrix during the 1st round of Delphi, and, based on it, the weighting for each potential indicator of the instrument, from the product of the average of the scores assigned to the safety dimension and the average of the scores assigned to efficacy dimension.

The three weighted composite indicators created were than analyzed for reliability through intra- and inter-rater agreement. In both cases, intraclass correlation coefficients (ICC) and Bland-Altman graphs were analyzed. In addition, histogram and box-plot graphs were calculated to analyze the usefulness of the indicators in identifying different levels of prescription quality and the difference between the three composite indicators [48, 49].

### Ethical aspects

This study was approved by the Ethics and Research Committee of the Hospital Universitário Onofre Lopes under n˚ 64367517.3.0000.5292.

## Results

### Face, content and criteria validity

Initially, 29 potential indicators were listed: number of drugs/prescription, absence of erasure, patient's full name, address, date of birth, prescriber's full name, identification, institution's name, institution's address, date of prescription, use of abbreviations, use of acronyms/symbols, active ingredient, drugs with similar names, concentration, frequency of administration,

record of allergy report, route of administration, dosage, pharmaceutical form, duration of treatment, directions on the use of the drug, non-pharmacological recommendations, medicine included in the institutional list officially approved, potentially dangerous drug, vague expression, drug with narrow therapeutic index, drug in therapeutic duplication and electronic prescription [26, 37–39].

Among the 15 experts who agreed to participate in Delphi were 6 pharmacists, 4 physicians, 3 nurses and 2 dentists. Of the pharmacists, four had dispensing experience in a hospital environment and two in an outpatient setting; meanwhile, three physicians had experience in prescribing in an outpatient setting and one had experience in a hospital setting. All three nurses had experience with care in both outpatient and hospital settings, while one dentist had experience in prescribing in an outpatient setting and the other in a hospital setting. In addition, everyone has experience of work processes in academia and the health services.

For the 1st round, nine indicators were agreed: prescriber identification, prescription date, concentration, dosage, pharmaceutical form, route of administration, frequency of administration, duration of treatment and directions on the use of the drug.

In the 2nd round, seven indicators were agreed upon: electronic prescription, absence of erasures, patient's full name, date of birth, record of allergy report, active ingredient and non-pharmacological recommendations.

In the 3rd round, 1 indicator was agreed upon (medicine included in the institutional list officially approved), ending the Delphi panel with 17 (57.1%) indicators agreed among the 29 proposed.

The criterion validity was analyzed after expert consensus, based on the level of evidence of the reference that supports the recommendation for each indicator, if the related practice has a proven relationship with the prevention of medication errors. The detailed results of the relevance and adequacy analyses in the three Delphi rounds are shown in S1 Table.

## Reliability and usefulness of indicators

The reliability assessment found inter-rater agreement for the 17 consensus indicators, whose kappa indexes ranged from 0.65 (95% CI 0.48–0.82) to 1.0 (95% CI 1.0–1.0) and overall agreement rates ranged from 87% to 100% (S2 Table).

The subsequent utility assessment resulted in the exclusion of four indicators, one because it did not apply to the reality of the local context (electronic prescription) and three because the prevalence of the two evaluators was 100% (absence of erasure, patient's full name and date of prescription), which indicates that these items do not contribute to compromising the quality of prescription writing, since prescribers have already incorporated these items into their professional practice (S2 Table).

Once the validation process was completed, the final version of the instrument consisted of 13 indicators. As an integral part of QualiPresc instrument, in order to contribute to the optimization of the reproducibility of the QualiPresc application process, definitions/clarifications of the validated indicators were generated according to S3 Table.

Graphical analysis via histograms and box-plots showed that the distribution of scores for the composite indicators had a variability similar to the normal distribution (Fig 2), revealing the usefulness of monitoring the instrument. In this variability, the indicator is sensitive to the detection of low, medium and high performance in efficacy, safety and quality.

Additionally, the inter-rater reliability of the composite indicators was analyzed through Bland-Altman agreement analysis (Fig 2). In addition to the image, the agreement is reinforced by the absence of statistically significant differences between the measures of the two evaluators (p = 0.622, p = 0.633 and p = 0.615, respectively), and none of these three composite indicators (the data) have a proportion bias relating to the distribution of evaluation differences

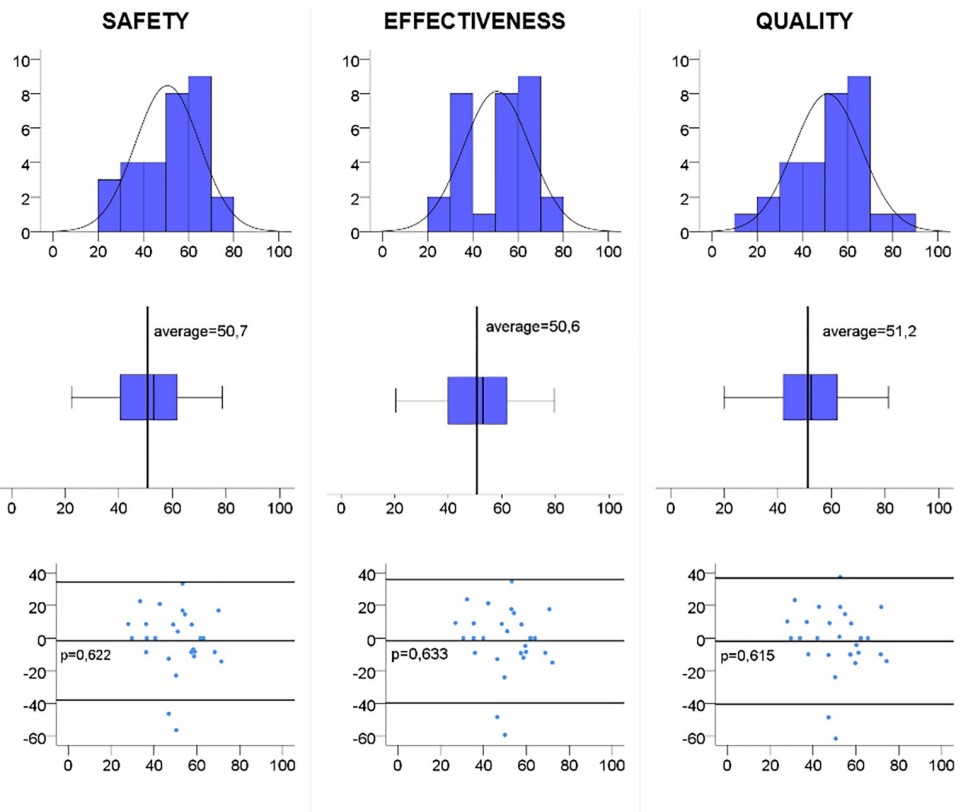

**Fig 2. Matrix-plot for reliability and utility analysis of composite indicators.** Thus, the Shapiro-Wilk test and intraclass correlation coefficient (ICC) test-retest were calculated from the quality indicator, resulting in p = 0.38 and ICC = 1.0, respectively.

among participants that is evenly distributed along the mean line and between the lower and upper limits of the 95% CI (p = 0.574, p = 0.574 and p = 0.525, respectively).

After transforming the three composite indicators to base 100 and comparing the means of each one, they are highly correlated (p<0.05), and the p-value of the difference between the means is similar (p>0.05) in a comparison of paired averages, showing that it is not useful to measure the three composite indicators because the quality is sufficient to represent the concept with the weight of each indicator. This can also be seen in Fig 2, which reveals similar data.

## QualiPresc instrument content

Once the validation process is complete, QualiPresc uses a concise template in Excel® format (S1–S3 Files). QualiPresc consists of 3 spreadsheets: the 1st is intended for the operational definition of indicators; the 2nd is for the tabulation of data collected from drug prescriptions; and the 3rd is for the evaluation and monitoring of the quality of these prescriptions, in general and by indicator, through trend graphs.

## Discussion

### General contributions

Drugs constitute one of the five foundations for quality care in a health system where accessibility, high quality and ordering primary care are three of the basic principles that are essential

for this to become a reality [41]. From this perspective, QualiPresc contributes to monitoring the quality of drug prescriptions at this level of health care, proposing to overcome the dangerous lack of validated instruments to assess the quality of prescription writing, based on structuring from the prescriber, the patient and the drug information.

Thus, this instrument can contribute to the WHO efforts to reduce medication errors by 50% over 5 years [38], both through research and through its use in health service regulation strategies or collaborative quality improvement projects [41]. This initiative aims to improve the quality of drug prescriptions in primary care, where medication errors are the main safety concern [2–5].

## Validity of QualiPresc in assessing the quality of prescriptions in primary care

Face and content validity were assessed using an internationally recognized consensus method [42], whose expert panel covered both a target-audience that measures quality and a target-audience that has its performance measured [40]. Additionally, since the kappa index has limitations [50], reliability was assessed in light of the prevalence of compliance, along with overall agreement. The sample size used in the expert committee and in the reliability pilot study are recognized as adequate in other studies [43, 51, 52].

Potential methodological challenges concerning the use of composite indicators were overcome through weighting by expert consensus, which is considered a valid option [46, 53]. In relation to a possible cut off point, as all indicators are considered essential, the quality goal is to reach 100%.

The validity of QualiPresc can also be verified by the justification of each indicator. For example, reference to the drug by trade name, as opposed to the "active ingredient", goes against the WHO's target of 100% of drugs being prescribed by generic name [37] in order to facilitate access and rational use of the drugs [38]. In relation to the indicator "medicine included in the institutional list officially approved", failure to comply with the list of essential drugs implies the unavailability of the prescribed drug in the public sector, which forces it to be obtained via the private sector. This does not always represent the best scientific evidence of safety and efficacy [54]. In addition, the omission of the "concentration" of the prescribed drug which consists of a numerical unit and a unit of measure, expressed via an abbreviation/acronym/symbol, can compromise the efficacy and safety of the drug treatment. As "concentration", "dosage", "frequency of administration" and "duration of treatment" are interdependent items, failing to indicate any of these may result in an underdose or overdose [55]. "Pharmaceutical forms" require specific "routes of administration", which are interdependent items that consider the convenience, stability, biological compatibility, speed of onset and duration of action and undesirable effects [55]. "Directions on the use of the drug" and "non-pharmacological recommendations" complement the other items mentioned above in order to contribute to the efficacy and safety of the pharmacological therapy [26, 56]. Additionally, "recording an allergy report" constitutes an impediment to the prescription of a drug to which the patient is allergic and cautions against other representatives of the same pharmacological class [56]. Furthermore, the "prescriber´s identification" enable contact with the prescriber should any clarification or pharmaceutical interventions be needed and the "patient's date of birth" constitutes an additional patient identifier [56].

Although valid and reliable, the electronic prescription indicator was not considered to be useful in the context of primary care, as aspects of infrastructure and professional qualification need to be fulfilled prior to its implementation. In Brazil, only 51.2% of Basic Health Units have a computer and only 35.4% have internet access [57]. There is a tendency for municipalities with low human development Index to have more precarious Basic Health Units, where

the most influential factor is the average municipal income. In this movement, the factor that negatively affects the infrastructure of Basic Health Units is that of being located in the North or Northeastern region, which constitutes very strong evidence of the regional inequality historically established in the country [58]. The evaluation of its effects in primary care found a reduction in prescribing errors when applied to a limited number of potentially dangerous drugs and with physician-pharmacist communication [59]. That said, electronic prescription can contribute to the reduction of prescription errors, as long as its implementation is properly planned and monitored, and its access authorized through prior training by specialists in patient safety [60, 61].

The potential readability indicator was excluded because it makes it impossible to apply the other indicators due to the impossibility of reading the prescription content. In addition, the readability test requires face-to-face administration of a questionnaire to at least 20 users, preferably from the target population of the drug, and excludes health professionals, in order to avoid results biased by their specialized knowledge [62, 63].

Although the panel of experts did not agree on the inclusion of potential indicators on the use of abbreviations/acronyms/symbols, when focusing on the structuring of the prescription document and the technical specifications of the drug, QualiPresc considers compliance with abbreviations/acronyms/symbols, as it imposes the mandatory presence of items such as active ingredient, dosage, concentration, route of administration, frequency of administration and duration of treatment. In addition, it considers that abbreviations/acronyms/symbols demonstrate errors related to the medication and are not accepted for compliance with the other indicators, in accordance with use/non-use recommendations [26, 64], as per S3 Table.

## Practical application of the QualiPresc

This set of validated indicators can be applied both by external evaluators (inspectors, accreditors, certifiers, auditors) and internal ones (self-assessment and continuous improvement of the health service), preferably pharmacists, who have a specific assignment to evaluate the prescription with regard to pharmacological, legal and biopsychosocial aspects [56]. In addition, it can be an instrument for teaching patient safety and good prescription practices in the academic environment.

QualiPresc is an alternative to long and paper-dependent measuring instruments and is freely available and easily used by health professionals from any primary care service, for the automatic collection and analysis of data monitoring. An average of 1 hour is invested to analyze 30 prescriptions.

It also constitutes software prototyping to be developed for the aggregate analysis of the quality of prescriptions in primary care networks, periodically, for application of the results in quality control charts, with a view to monitoring and continuous improvement, which will be developed initially, in the same municipality where the reliability and usefulness of the potential QualiPresc indicators were analyzed.

## Study limitations

Convergent validity was not assessed due to the absence of instruments to assess the quality of prescriptions aimed at observing prescription writing, which constitute the gold standard for this comparison.

Criteria validity was evaluated based on the references of each indicator, but none represents strong recommendations, despite the very high consensus on face and content validity. However, the availability of QualiPresc will favor the production of robust evidence for the criteria validity of the prescription writing when relating them to medication errors.

QualiPresc has been validated for use in the context of primary care in a country with continental dimensions, whereas the reality is very diverse. The application of QualiPresc in significantly different contexts, such as secondary care and tertiary care, requires cross-cultural adaptation and new content validity, for which the list of 29 potential indicators may be useful.

## Conclusion

The study shows the validity, reliability and usefulness of QualiPresc for the assessment of prescription writing, in the scope of primary care. Application to significantly different contexts, such as secondary care and tertiary care, requires cross-cultural adaptation and new content validity. Educators, managers and healthcare professionals can access it free of charge online to monitor and provide feedback to drug prescribers.

## Supporting information

**S1 Table. Validity analysis of potential indicators.**
(DOC)

**S2 Table. Reliability and utility analysis of consensual indicators.**
(DOC)

**S3 Table. Definitions and clarifications of the validated indicators.**
(DOC)

**S1 File. Portuguese QualiPresc.**
(PDF)

**S2 File. English QualiPresc.**
(PDF)

**S3 File. Spanish QualiPresc.**
(PDF)

## Author Contributions

**Conceptualization:** Almária Mariz Batista, Zenewton André da Silva Gama.

**Data curation:** Almária Mariz Batista.

**Formal analysis:** Almária Mariz Batista, Zenewton André da Silva Gama.

**Investigation:** Almária Mariz Batista.

**Methodology:** Almária Mariz Batista, Zenewton André da Silva Gama.

**Project administration:** Almária Mariz Batista, Zenewton André da Silva Gama.

**Supervision:** Zenewton André da Silva Gama.

**Validation:** Zenewton André da Silva Gama, Dyego Souza.

**Visualization:** Almária Mariz Batista, Zenewton André da Silva Gama, Dyego Souza.

**Writing – original draft:** Almária Mariz Batista.

**Writing – review & editing:** Almária Mariz Batista, Zenewton André da Silva Gama, Dyego Souza.

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
