## [Decision Letter · Decision Letter 0]

25 Jan 2022

PONE-D-21-37829VALIDATION OF THE QUALIPRESC INSTRUMENT FOR ASSESSING THE QUALITY OF THE DRUGS PRESCRIPTION UNDER WRITING CRITERIA IN THE PRIMARY HEALTH CAREPLOS ONE

Dear Dr. Mariz Batista,

Thank you for submitting your manuscript to PLOS ONE. After careful consideration, we feel that it has merit but does not fully meet PLOS ONE’s publication criteria as it currently stands. Therefore, we invite you to submit a revised version of the manuscript that addresses the points raised during the review process.

We look forward to receiving your revised manuscript.

Kind regards,

Benito Soto-Blanco, DVM, MSc, PhD

Academic Editor

PLOS ONE

Journal Requirements:

2. Please include captions for your Supporting Information files (Portuguese qualipresc, English qualipresc, and Spanish qualipresc) at the end of your manuscript, and update any in-text citations to match accordingly. Please see our Supporting Information guidelines for more information: http://journals.plos.org/plosone/s/supporting-information. 

Reviewers' comments:

Reviewer's Responses to Questions

**Comments to the Author**

1. Is the manuscript technically sound, and do the data support the conclusions?

Reviewer #1: Partly

Reviewer #2: Yes

2. Has the statistical analysis been performed appropriately and rigorously? 

Reviewer #1: Yes

Reviewer #2: Yes

3. Have the authors made all data underlying the findings in their manuscript fully available?

Reviewer #1: Yes

Reviewer #2: Yes

4. Is the manuscript presented in an intelligible fashion and written in standard English?

Reviewer #1: No

Reviewer #2: Yes

5. Review Comments to the Author

Reviewer #1: This is a study whose novelty is relatively clear, but there were some points which makes this manuscript difficult to understand.

1. (Abstract, line 28) What “writing criteria” mean? Is this a commonly used phrase?

2. (Abstract, Methods) Please write about where this study was taken place, who are the target patients or medical institutes of the study, and their numbers.

3. (Methods, line 105) How did you create QualiPresc ? Does “Step1. Validity analysis” correspond to the method of creating QualiPresc ? In that case, it is bizarre that how the QualiPresc was created is written in the section of validation analysis. In other words, you should write separately about development and validation of QualiPresc.

4. (Results) All the Figures are blurred, and difficult to see. Please increase the quality of the Figures. You should check it before submission. In addition, Figures 2-4 can be presented by Tables. Why did you present them in Figures ?

5. (Results, Figure 4) Does Figure 4 corresponds to the finalized QualiPresc ? Please write more about how to use it in practice.

6. (Discussion) As you write in the limitations, the generalizability of QualiPresc may be limited. For example, it might be good to use it in other countries. Therefore, you should write about where this method can be applicable in the Conclusions and Conclusions section of Abstract.

Reviewer #2: Dear Authors,

I have enjoyed reading your manuscript. I commend you for doing this research to validate this instrument which will definitely improve patient safety.

Below are some suggestions/areas of improvement to consider.

Abstract: Objective, line 29-31: Please the sentence is too long, making it difficult to retain the main message from the beginning to the end of the sentence. Consider dividing into at least two sentences so that it is easier to comprehend.

Abstract: Line 36 & 38: Consider spelling the numbers "29" and "12" since they are at the beginning of the sentences.

Introduction: Line 52: What is the "damage" in the sentence referring to? Is it injury or harm related to unsafe medication process? It is not clear to me.

Introduction: Line 56: Is this "suffer harm" related to general harm from healthcare or medication-related harm?

Introduction: Line 67: "In this context .." What is the context? Please elaborate.

Introduction: Line 67-68: "technological advances are fast ...". How does the fast technological advances lead to less adequate prescription practices? It is not clear to me.

Introduction: Line 72: "Writing criteria". Please consider briefly expanding what the writing criteria is in the context of primary care practice in Brazil. What should be the ideal practice for the writing criteria and why does the current practice warrant the need for assessing the quality of the prescription with the instrument?

Introduction: Line 74: Not very clear to me what "the care of" means in the sentence.

Introduction: Line 83-85: Could the use of electronic prescribing (e.g., via the use of an electronic health record system with prescription templates) in well resourced settings contribute to the reason why there is there is "no valid, reliable and useful instrument that allows for the assessment of prescription quality in terms of writing criteria"?

Introduction: Line 87: "results in ignorance about the impacts of these aspects on medication errors." Please can you briefly elaborate on the impact (e.g., patient harm or cost related to harm to patients) of the absence of a tool to assess prescription writing criteria on patient safety if there is literature available?

Methods: Line 117: How did you select which specialists you called over the phone for recruitment?

Methods: Line 118: "systematic of the study". Please it is not clear what this phrase means in the sentence. May need to rephrase to make the meaning clearer.

Methods: Line 119: "Informed Consent Form". Please consider using lower case letters at the beginning of each word hence 'informed consent form'.

Methods: Line 120. "Delphi Panel". Consider small letter at the beginning, hence 'panel'.

Methods: Line 120: "3 rounds". Consider spelling out 3 since it is a figure less than 10.

Methods: Line 142-146: The sentence is too long. Makes it difficult to read and understand. Please consider dividing into a minimum of 2 sentences.

General comment for the methods section: Overall, in the methods section, some of the paragraphs are very short (just 2 sentences) and it may appropriate to combine some paragraphs to reduce the total number of paragraphs.

Methods: Line 165: "If the response it yes, then ok." Please check and revise. Not clear to me.

Methods: Line 172: "Weighing". Please check if you intended to write 'weighting'. If not, then consider checking if it will be appropriate.

Results: Lines 232 and 233: "is completed" and "consists". Please check if you need to use the past tense since it has happened.

Discussion: Line 309: "UBS": Please the international readers may not understand the abbreviation. Consider writing out.

Thank you.

6. PLOS authors have the option to publish the peer review history of their article (what does this mean?). If published, this will include your full peer review and any attached files.

Reviewer #1: No

Reviewer #2: No

---

## [Author Response · Author response to Decision Letter 0]

16 Feb 2022

Journal Requirements / Authors' Response

1. This request has been fulfilled in accordance with https://journals.plos.org/plosone/s/file?id=wjVg/PLOSOne_formatting_sample_main_body.pdf and https://journals.plos.org/plosone/s/file?id=ba62/PLOSOne_formatting_sample_title_authors_affiliations.pdf

2. Please include captions for your supporting information files (Portuguese qualipresc, English qualipresc, and Spanish qualipresc) at the end of your manuscript, and update any in-text citations to match accordingly. Please see our Supporting Information guidelines for more information: http://journals.plos.org/plosone/s/supporting-information. 

2. This request has been fulfilled in accordance with the Supporting Information guidelines http://journals.plos.org/plosone/s/supporting-information.

This information has been added to the Discussion, line 362 and to the Supporting information, lines 597-599 of the manuscript.

3. Please review your reference list to ensure that it is complete and correct. If you have cited papers that have been retracted, please include the rationale for doing so in the manuscript text, or remove these references and replace them with relevant current references. Any changes to the reference list should be mentioned in the rebuttal letter that accompanies your revised manuscript. If you need to cite a retracted article, indicate the article’s retracted status in the reference list and also include a citation and full reference for the retraction notice. 

3. References 17-25 and 36 have been added to the manuscript (lines 81 and 99, respectively) and can be found in the References section, lines 440-467 and 497-500, respectively

Reviewer #1 / Authors' Response 

1. (Abstract, line 28) What does “writing criteria” mean? Is this a commonly used phrase? By “writing criteria”, we mean prescription writing, which covers the prescriber, patient and drug information. This has now been clarified in line 28 of the Abstract, and the expression “writing criteria” has been replaced by the term “prescription writing” in the title and throughout the body of the article.

2. (Abstract, Methods) Please write about where this study has taken place, who are the target patients or medical institutes of the study, and their numbers. 

The study was conducted in a municipality in Northeastern Brazil, based on prescriptions prepared in January 2021 by physicians assigned to 18 Basic Health Units and filed in 6 distribution/dispensing units. This information has been added to the Abstract (Methodology), lines 32-34.

3. (Methods, line 105) How did you create QualiPresc? Does “Step 1. Validity analysis” correspond to the method of creating QualiPresc? In that case, it is bizarre that how the QualiPresc was created is written in the section of validation analysis. In other words, you should write separately about the development and validation of QualiPresc. 

No, the validity analysis does not correspond to the method used to create QualiPresc. The validity analysis covers analysis of the relevance (face validity), adequacy (content validity), and theoretical support (criteria validity) of each potential indicator used to assess prescription quality. In this case, the relevance indicates how important the potential indicator is in assessing the quality of the prescription, the adequacy indicates how much the potential indicator is able to measure the quality of the prescription, and the theoretical support refers to the literature on which the potential indicator is based. This information has been added to the Methods section of the manuscript, lines 123-128.

4. (Results) All the Figures are blurred, and difficult to see. Please increase the quality of the Figures. You should check it before submission. In addition, Figures 2-4 can be presented by Tables. Why did you present them in Figures? 

We have made changes to the resolution of Figures 1 and 5 (resolution increased from 300 dpi to 600 dpi), and Figures 2-4 have now been converted to Tables (Tables 1-3), as requested. Thus, Figure 5 was renamed Figure 2 in the revised manuscript.

5. (Results, Figure 4) Does Figure 4 correspond to the finalized QualiPresc? Please write more about how to use it in practice. 

No, Figure 4 (renamed Table 3 in the revised manuscript) does not correspond to the finalized QualiPresc, but is an integral part of the QualiPresc instrument. This has been clarified in the Results section of the manuscript, in line 251 (“As an integral part of QualiPresc instrument,”). QualiPresc consists of 3 spreadsheets: the 1st is intended for the operational definition of indicators; the 2nd is for the tabulation of data collected from the drug prescriptions; and the 3rd is for the evaluation and monitoring of the quality of these prescriptions, in general and by indicator, through trend graphs. This information has been added to the Discussion section of the manuscript, lines 365-369.

6. (Discussion) As you write in the limitations, the generalizability of QualiPresc may be limited. For example, it might be good to use it in other countries. Therefore, you should write about where this method can be applicable in the Conclusions and the Conclusions section of the Abstract. 

The application of QualiPresc in significantly different contexts, such as secondary care and tertiary care, requires cross-cultural adaptation and new content validity, for which the list of 29 potential indicators may be useful. This information has been added to the manuscript, in the Discussion, lines 385-388; the Conclusion, lines 391-393; and the Abstract (Conclusion), lines 46-47.

Reviewer #2 / Authors' Response 

Abstract: Objective, line 29-31: The sentence is too long, making it difficult to retain the main message from the beginning to the end of the sentence. Consider dividing into at least two sentences so that it is easier to comprehend. 

We have changed this sentence in accordance with Proofreading.

Abstract: Line 36 & 38: Consider spelling the numbers "29" and "12", since they are at the beginning of the sentences. 

We have changed this in accordance with your request (lines 39-40).

Introduction: Line 52: What is the "damage" in the sentence referring to? Is it injury or harm related to an unsafe medication process? It is not clear to me. 

This term concerns the harm related to health care. This has now been clarified in the Introduction, line 55.

Introduction: Line 56: Is this "suffer harm" related to general harm from healthcare or medication-related harm? 

It is harm related to health care. This has now been clarified in the Introduction, line 59.

Introduction: Line 67: "In this context .." What is the context? Please elaborate. 

The context is primary care, and this has now been clarified in the Introduction, line 70.

Introduction: Line 67-68: "technological advances are fast ...". How does the fast technological advances lead to less adequate prescription practices? It is not clear to me. Fast technological advances demand commitment and time investment from prescribers in order to gain the appropriate knowledge inherent to this health technology, including the best scientific evidence about efficacy, safety and cost-effectiveness. This has now been clarified in the Introduction, lines 72-74.

Introduction: Line 72: "Writing criteria". Please consider briefly expanding what the writing criteria is in the context of primary care practice in Brazil. What should be the ideal practice for the writing criteria and why does the current practice warrant the need for assessing the quality of the prescription with the instrument? 

Ideal prescription writing encompasses the prescriber, the patient, and the drug information; however, this information is often missing or incomplete. In Brazil, the practice of prescription writing is based on recommendations from the Ministry of Health. This information has now been added to the Introduction, lines 79-82. Additionally, the expression “writing criteria” has been replaced by the expression “prescription writing”, in the title and throughout the body of the article.

Introduction: Line 74: Not very clear to me what "the care of" means in the sentence. 

The expression “the care of” has been replaced by the word “across” in the Introduction, line 83.

Introduction: Line 83-85: Could the use of electronic prescribing (e.g., via the use of an electronic health record system with prescription templates) in well-resourced settings contribute to the reason why there is "no valid, reliable and useful instrument that allows for the assessment of prescription quality in terms of writing criteria"? 

The sentence “Furthermore, this cannot be resolved through an electronic prescription system, even in well-resourced settings.” has now been added to the Introduction, lines 98-99.

Introduction: Line 87: "results in ignorance about the impacts of these aspects on medication errors." Please can you briefly elaborate on the impact (e.g., patient harm or cost related to harm to patients) of the absence of a tool to assess prescription writing criteria on patient safety, if there is literature available? 

The phrase “given that medication errors represent the main safety incidents resulting in adverse events in primary care” has now been added to the Introduction, lines 96-98.

Methods: Line 117: How did you select which specialists you called over the phone for recruitment? 

The following text has been added to the Methods section, lines 134-137: “Pharmacists, physicians, nurses, and dentists with experience of work processes in academia and the health service were invited to participate in the Delphi panel. Thus, it covered a target audience that either evaluates the quality of the prescription and/or has its performance evaluated. Among the 21 invited participants, 15 agreed to participate.” The invited participants were considered to be key informants who met the inclusion criteria.

Methods: Line 118: "systematic of the study". Please it is not clear what this phrase means in the sentence. May need to rephrase to make the meaning clearer. 

This sentence has been reworded to clarify that this relates to the methodology of the Delphi panel (see the Methods section, lines 138-139). “The recruitment of specialists took place via telephone contact, during which the methodology of the Delphi panel was presented.”

Methods: Line 119: "Informed Consent Form". Please consider using lower case letters at the beginning of each word hence 'informed consent form'. 

We have changed this in accordance with your request (line 140).

Methods: Line 120. "Delphi Panel". Consider small letter at the beginning, hence 'panel'. 

We have changed this in accordance with your request (line 141).

Methods: Line 120: "3 rounds". Consider spelling out 3 since it is a figure less than 10. 

We have changed this in accordance with your request (line 141).

Methods: Line 142-146: The sentence is too long. Makes it difficult to read and understand. Please consider dividing into a minimum of 2 sentences. 

We have changed this in accordance with your request (line 160). 

General comment for the methods section: Overall, in the methods section, some of the paragraphs are very short (just 2 sentences) and it may appropriate to combine some paragraphs to reduce the total number of paragraphs. 

We have changed this in accordance with your request. 

Methods: Line 165: "If the response it yes, then ok." Please check and revise. Not clear to me. 

The phrase “If the response it yes, then ok” has been replaced by the text “Where there was consensus about the usefulness of the indicator, it was considered useful; otherwise, it was excluded” in the Methods section, lines 181-182.

Methods: Line 172: "Weighing". Please check if you intended to write 'weighting'. If not, then consider checking if it will be appropriate. 

We have changed this in accordance with your request (lines 189, 194 & 300). 

Results: Lines 232 and 233: "is completed" and "consists". Please check if you need to use the past tense since it has happened. 

We have changed this in accordance with your request (lines 250-251). 

Discussion: Line 309: "UBS": Please the international readers may not understand the abbreviation. Consider writing out. 

We have changed this in accordance with your request throughout the body of the article.

---

## [Decision Letter · Decision Letter 1]

22 Mar 2022

PONE-D-21-37829R1VALIDATION OF THE QUALIPRESC INSTRUMENT FOR ASSESSING THE QUALITY OF DRUG PRESCRIPTION WRITING IN PRIMARY HEALTH CAREPLOS ONE

Dear Dr. Mariz Batista,

Thank you for submitting your manuscript to PLOS ONE. After careful consideration, we feel that it has merit but does not fully meet PLOS ONE’s publication criteria as it currently stands. Therefore, we invite you to submit a revised version of the manuscript that addresses the points raised during the review process.

We look forward to receiving your revised manuscript.

Kind regards,

Benito Soto-Blanco, DVM, MSc, PhD

Academic Editor

PLOS ONE

Journal Requirements:

Reviewers' comments:

Reviewer's Responses to Questions

**Comments to the Author**

1. If the authors have adequately addressed your comments raised in a previous round of review and you feel that this manuscript is now acceptable for publication, you may indicate that here to bypass the “Comments to the Author” section, enter your conflict of interest statement in the “Confidential to Editor” section, and submit your "Accept" recommendation.

Reviewer #1: All comments have been addressed

Reviewer #2: All comments have been addressed

2. Is the manuscript technically sound, and do the data support the conclusions?

Reviewer #1: Yes

Reviewer #2: Yes

3. Has the statistical analysis been performed appropriately and rigorously? 

Reviewer #1: Yes

Reviewer #2: Yes

4. Have the authors made all data underlying the findings in their manuscript fully available?

Reviewer #1: Yes

Reviewer #2: Yes

5. Is the manuscript presented in an intelligible fashion and written in standard English?

Reviewer #1: Yes

Reviewer #2: Yes

6. Review Comments to the Author

Reviewer #1: Thank you for the revision. I have a few comments.

1. (Tables) I think Tables 1-3 need to be shown in the main manuscript, not in supplementary information.

2. (Discussion, lines 363) It is bizarre that the contents of QualiPresc are shown for the first time in Discussion. You should indicate it in the Methods or Results. In addition, QualiPresc in English version needs to be written in English, but not in another language.

Reviewer #2: Dear Authors,

Thank you for considering my feedback in the revised manuscript. You have thoroughly addressed all my concerns in your revision.

Thank you.

Reviewer #2

7. PLOS authors have the option to publish the peer review history of their article (what does this mean?). If published, this will include your full peer review and any attached files.

Reviewer #1: No

Reviewer #2: **Yes: **Yaw B Owusu

---

## [Author Response · Author response to Decision Letter 1]

11 Apr 2022

Journal Requirements / Authors' Response

None of the papers cited in this manuscript were retracted.

Reviewer #1 

6. Review Comments to the Author

1. (Tables) I think Tables 1-3 need to be shown in the main manuscript, not in supplementary information.

1. We agree. Tables 1-3 will be in the manuscript. They are not Supplementary Information. The current view at the end of the manuscript is only during the review process, as this is Plos One's organization during this process (tables and figures). We have included the titles of all tables and figures in the text to indicate the approximate location where they will be inserted. See manuscript template (Download sample manuscript body (PDF)).

2. (Discussion, lines 363) It is bizarre that the contents of QualiPresc are shown for the first time in Discussion. You should indicate it in the Methods or Results. 

2. Changes made. This information is presented in the Results topic (lines 277-282).

3. In addition, QualiPresc in English version needs to be written in English, but not in another language. 

3. Changes made. View Supplementary Files 1-3.

---

## [Editor Report · Decision Letter 2]

14 Apr 2022

VALIDATION OF THE QUALIPRESC INSTRUMENT FOR ASSESSING THE QUALITY OF DRUG PRESCRIPTION WRITING IN PRIMARY HEALTH CARE

PONE-D-21-37829R2

Dear Dr. Mariz Batista,

We’re pleased to inform you that your manuscript has been judged scientifically suitable for publication and will be formally accepted for publication once it meets all outstanding technical requirements.

Kind regards,

Benito Soto-Blanco, DVM, MSc, PhD

Academic Editor

PLOS ONE

---

## [Editor Report · Acceptance letter]

22 Apr 2022

PONE-D-21-37829R2 

VALIDATION OF THE QUALIPRESC INSTRUMENT FOR ASSESSING THE QUALITY OF DRUG PRESCRIPTION WRITING IN PRIMARY HEALTH CARE 

Dear Dr. Mariz Batista:

I'm pleased to inform you that your manuscript has been deemed suitable for publication in PLOS ONE. Congratulations! Your manuscript is now with our production department. 

Kind regards, 

on behalf of

Dr. Benito Soto-Blanco 

Academic Editor

PLOS ONE